# Developing outcome measures assessing wound management and patient experience: a mixed methods study

The Bluebelle Study Group

Population Health Sciences, Bristol Medical School, University of Bristol, Canynge Hall, 39 Whatley Road, Bristol, BS8 2PS

**Correspondence to**
Dr Daisy Elliott;
daisy.elliott@bristol.ac.uk

## ABSTRACT

**Objectives** To develop outcome measures to assess practical management of primary surgical wounds and patient experience.

**Design** Mixed methods, including qualitative interviews and data extraction from published randomised controlled trials (RCTs).

**Setting** Two university-teaching NHS hospitals and three district NHS hospitals in the South West and Midlands regions of England.

**Participants** Sixty-four patients and 15 healthcare professionals from abdominal general surgical specialities and obstetrics (caesarean section).

**Methods** Measures were developed according to standard guidelines to identify issues relevant to patients' experiences of surgical wounds and dressings, including analysis of existing RCT outcomes and semi-structured interviews. These were written into provisional questionnaire items for a single outcome measure. Cognitive interviews with patients and healthcare professionals assessed face validity, acceptability and relevance. Findings from interviews were regularly shared with the study team who suggested amendments to modify and reword items to improve understanding before further iterative testing with patients and healthcare professionals.

**Results** Analyses of existing RCT outcomes and interviews produced a total of 69 issues. Pretesting and iterative revision established the need for two separate measures. One measure addresses healthcare professionals' experience of wound management in two key areas: exudate and its impact, and allergic reactions to the dressing. The other measure addresses patients' experience of wounds in seven key areas: wound comfort, dressing removal, dressings to protect the wound, impact on daily activities, ease of movement, anxiety about the wound and satisfaction with dressing. Each measure took less than five min to complete and both were understood and acceptable to patients and healthcare professionals.

**Conclusion** This in-depth study has developed two measures to assess practical management of primary surgical wounds and patient experience. Further work to test their validity, reliability and application to other settings is now required.

**Trial registration number** HTA - 12/200/04; Pre-results.

## INTRODUCTION

An estimated 234 million major surgical procedures are undertaken worldwide every year.[1] It is a common practice to apply dressings over the closed wound in adult surgery

### Strengths and limitations of this study

► This is the first study to explore the important issues related to the practical management of primary surgical wounds and patient experience immediately following surgery.
► This study used robust methods to identify key issues of outcome that could be used to inform decision making around dressings. Interviews provided a rich account of the key factors that affected wound management and patient experience while a purposeful sampling strategy ensured that perspectives were captured from a range of participants. Data produced from the interviews were supplemented by analysis of existing randomised controlled trials (RCT) outcomes to ensure a comprehensive list of issues was considered.
► Future work is needed to test the reliability, validity and sensitivity of the new measures.

and many different dressing types are available.[2] A recent Cochrane systematic review summarised data relating to wound dressings and risk of surgical site infection (SSI) in primary surgical wounds. No evidence was found to suggest that any type of dressing significantly reduced the risk of developing an SSI compared with leaving wounds exposed; neither was there any benefit associated with particular dressings.[3]

Decision making around dressings may therefore need to be informed by other properties and qualities that dressings can offer, such as absorption of exudate, patient comfort, offering physical protection, facilitating wound observation and meeting patients' desires for wound coverage.[4] While measures for assessment of wound cosmesis (in the longer term) are available,[5 6] there is a lack of well-developed and validated measurement tools relating to practical wound management or patient experience.[4 7 8] Such an instrument could be used to monitor the care of individual patients (eg, assessing the ability of dressings to manage specific

symptoms), audits (eg, quality assurance) and research (eg, comparing patient satisfaction).

The development of patient-reported outcome measures (PROMs) increasingly includes the use of qualitative research methods which provide the opportunity to elicit and characterise patients' experiences of their health conditions and treatment.[9][10] Qualitative methods can also define health professionals' experiences of care and management.[11] Data can be supplemented by expert input and studies in published literature.[12][13] This article describes the development of measures to assess practical wound management issues and patient experience associated with primary surgical wounds.

## METHODS
### Study design
Measures were developed according to an existing framework for developing PROMs,[14][15] also incorporating guidance on eliciting health domain concepts using qualitative methodologies.[12][13][16] The study is reported according to qualitative reporting guidelines (see online supplementary file 1). Phase 1 aimed to produce a comprehensive list of potential issues relating to wound and dressing experience and practical management issues. Phase 2 developed issues identified from phase 1 into questionnaire items. Phase 3 evaluated the measures for acceptability and relevance using cognitive interviews with patients and healthcare professionals. The final part of development (phase 4) consisted of psychometric testing and will be reported elsewhere. Written informed consent was provided by all participants.

### Phase 1: Generation of relevant issues
#### Interviews
Interviews were conducted with patients to explore and characterise experiences of wounds and dressings. Participants were recruited as part of a wider feasibility study to explore whether a trial comparing different types of dressings, and dressing versus no dressings, is possible (The Bluebelle study: a feasibility study of three wound dressing strategies in elective and unplanned surgery, HTA - 12/200/04[2][17]). Participants were recruited from two University-teaching NHS hospitals and three district NHS hospitals in the South West and Midlands regions of England. Eligible patients who had undergone, or were scheduled to undergo, an abdominal surgical procedure or caesarean section were identified and approached by research nurses and surgical trainees. The qualitative team contacted interested patients to arrange interviews. A purposeful sampling strategy ensured that perspectives were captured from a range of participants.[13] Within this sampling approach, maximum variation was sought in relation to age, gender, ethnicity, type of surgery, dressing type and location. A topic guide was developed (based on the literature and views of healthcare professionals in the Bluebelle study team) to ensure that discussions covered the same core issues but with sufficient flexibility to allow

new issues of importance to the participants to emerge (see online supplementary file 2).

Interviews were audio-recorded and transcribed in full. Transcripts were imported into NVivo (version 10). All data relating to post-surgical issues were assigned labels (coded) by two experienced qualitative researchers. Data were analysed using techniques of constant comparison derived from grounded theory methodology, and emerging codes across the dataset were then compared to look for shared or disparate views among participants.[18] A subset of approximately half of the interviews (n=19) was double coded by a third experienced researcher to highlight any differences in the interpretation of codes.[12] Data collection and analysis continued until the team were confident that saturation had been reached (ie, at which no more patterns or themes emerged from the data).[19]

#### Extraction of information from three systematic reviews
Systematic reviews were purposely selected to identify randomised controlled trials (RCTs) measuring outcomes relating to patient experience and management of wound healing. Since the wider Bluebelle study explored whether a trial comparing different types of dressings (including dressings versus the novel use of tissue glue as a dressing) with no dressing was possible, we selected three recent systematic reviews[4][20][21] to identify RCTs which included outcomes relevant to both dressings and the use of tissue adhesive. Although not published at the time of conducting this work, additional references from an updated version of one of the systematic reviews[7] were also provided. Published papers reporting the RCTs included in the systematic reviews were obtained where possible. Relevant data from the RCT reports were then extracted on the outcome (as described by the authors), the verbatim wording to measure outcome, who reported the outcome, the measurement scale and the assessment time point. Attempts were made to contact the authors for more information.

#### Synthesis of findings from interviews and data extraction
Identified issues were collated into an item tracking matrix, in line with guidance for developing PROMs.[22] This is available in the online appendix (see online supplementary file 3). The study team agreed on a set of words or phrases to reflect each issue and also noted additional phrasing made by participants in a subsequent column.[12] Issues which were conceptually similar were organised into categories. For instance, issues such as 'itchiness/irritation', 'presence of pulling sensation' and 'tightness of wound' were mapped into a 'wound comfort' category.

### Phase 2: 'Operationalisation': construction of a provisional measure
The item tracking matrix was used to determine which issues should be written into questionnaire items. Items featured words and phrases used by patients in the interviews to enhance content validity.[13][23]

### Phase 3: Pretesting
Participants were recruited from two University-teaching NHS hospitals in the South West and Midlands regions

of England. Patients who had undergone abdominal general surgery or caesarean section, as well as healthcare professionals involved in their postsurgical care, were approached. As in phase 1, sampling was purposeful to achieve maximum variation in relation to clinical role, age, gender and geographic location (for healthcare professionals) and age, gender, ethnicity, type of surgery, dressing type and location (for patients).

Cognitive interviews are used widely in questionnaire development[12] and involve asking respondents to verbalise their thoughts while answering questions.[24] This methodology enabled us to explore the acceptability of the measure and coverage of patients and healthcare professionals' concerns (in terms of language, accuracy, relevance, and layout[13]). During each interview, participants were asked to complete the measure by reading each item aloud and commenting on their understanding. Interviews were guided by a series of probes (eg, 'What does this item mean to you?', 'Are there other ways you would describe it?'[24]). Participants' body language (such as nodding or frowning) was also observed and prompted further discussion about specific items.[12] A copy of the topic guide is available (see online supplementary file 2).

The qualitative team maintained detailed field notes from each interview, describing suggestions for modifications and improvements to the provisional measures. Operationalisation and modification of the measures was an iterative process. Findings from the interviews and suggestions for amendments were regularly disseminated to the Bluebelle Study Group, which consisted of a multidisciplinary group of healthcare professionals, including surgeons, health services researchers and research nurses. Each stage of feedback informed amendments to modify and reword items to improve understanding, which was repeated following efforts to revise questions and eliminate problems.[24] This process continued until no new issues were identified and no further refinements were believed to be necessary.

## RESULTS
### Phase 1: Generation of relevant issues
#### Interviews
A total of 39 interviews were conducted between July 2014 and July 2015. Interviews were conducted in person (n=10), unless patients preferred to be interviewed via telephone (n=29). Interviews lasted an average of 25 min (range=15–50 min). The sample consisted of 27 women and 12 men, who mostly described themselves as white British (90%). They had a mean age of 56 years (range 22–88 years). Thirty seven of the 39 participants had either undergone abdominal general surgery (85%) or a caesarean section (15%), with an average of 18 days since their surgery (range=6–40 days). Two of the 39 patients were scheduled to undergo abdominal general surgery and discussed issues that they anticipated would be important to them. Participant demographics for phase 1 interviews are shown in table 1.

**Table 1** Participants' demographic details

| | | Phase 1 | Phase 3 | |
| | | Generation of relevant issues | Pretesting | |
|---|---|---|---|---|
| Qualitative interviews (n=79) | | 39 patients | 25 patients | 15 healthcare professionals |
| Age (years) | Range | 22–88 | 19–76 | 23–60 |
| | Mean | 56 | 54 | 41 |
| Sex | Female | 27 | 12 | 13 |
| | Male | 12 | 13 | 2 |
| Ethnicity | White | 35 | 22 | 14 |
| | Asian | 1 | 1 | 0 |
| | African | 2 | 1 | 1 |
| | Indian | 1 | 0 | 0 |
| | Filipino | 0 | 1 | 0 |
| Type of surgery | Abdominal | 33 | 25 | 15 |
| | Obstetric | 6 | 0 | 0 |
| Dressing type | Tissue adhesive | 7 | 5 | – |
| | Adhesive | 32 | 18 | – |
| | No dressing | 0 | 2 | – |
| Location | South West | 28 | 15 | 9 |
| | West Midlands | 11 | 10 | 6 |

### Extraction of information from three systematic reviews
Published papers for 26 studies that included outcomes relating to patient experience and management of wound healing were identified from the three systematic reviews.[25–50] Only two studies included a validated instrument, or modification of a validated instrument, to assess outcomes.[27 41] These were for long-term scarring and cosmesis.[5 6] However, no studies reported using validated measures relating to issues associated with practical wound management and patient experiences in the early postoperative period. Descriptions of outcomes were heterogeneous and often poorly defined. The most common reported outcomes related broadly to cosmetic result (reported in 15/26 studies), dressing changes (eg, frequency, comfort, ease of application and removal; reported in 11/26 studies) and skin reactions (eg, itching, blistering; reported in 10/26 studies). Full data extraction from the 26 studies is included in online supplementary file 4.

### Synthesis of findings from interviews and data extraction
When describing their experiences in the interviews, patients commented on several factors that affected perceptions of how well their wound was healing, including how it felt (tightness, pain and itchiness) and whether any fluid had leaked from the wound. Analysis

**Table 2** Categories identified

| Category | Example quote |
| --- | --- |
| Wound comfort | 'I've now got really itchy where the plaster goes. Which is uncomfortable.' (Patient, adhesive dressing) |
| Exudate and its impact | 'If I walked around it would get really damp. I mean it would soak my pyjamas and drip down my legs. It was quite manky really…Then they would put a sort of big, well, like a big plaster on top of that, and then they put a kind of absorbent pad over that, to absorb some of that liquid.' (Patient, adhesive dressing) |
| Reactions to the dressing | 'I was allergic to the surgical tape.' (Patient, adhesive dressing) |
| Dressing removal | 'I just completely soaked it [adhesive dressing] in the shower then my husband just took it off for me. But it was, it was really easy. Much easier than I thought.' (Patient, adhesive dressing) |
| Wound protection | 'I'd be worried about catching it [the wound], knocking it, or something getting in so that it became infected.' (Patient, adhesive dressing) |
| Impact on daily activities | 'With the glue [dressing] it's easy to shower. With a [adhesive] dressing it wouldn't be so easy to shower and you'd be worried.' (Patient, tissue adhesive dressing) |
| Ease of movement | What I do find is the dressings are a bit constricting, especially as I get a bit better because they don't turn with your body so easily and then I feel that it makes me feel more constricted.' (Patient, tissue adhesive dressing) |
| Anxiety about the wound | 'You could catch things just from the air. That made me think, 'Well, you'd need something to kind of protect it.' (Patient, adhesive dressing) |
| Satisfaction with dressing | 'Glue [as a dressing] requires no maintenance. I was very pleased. You don't have to change it you just leave it alone … I think that helps with the healing process physically and mentally.' (Patient, tissue adhesive dressing) |
| Wound appearance | 'If it was red and inflamed I would have thought, 'Something has gone wrong with it.'' (Patient, adhesive dressing) |

of existing RCT outcomes showed these issues had been captured in some previous (unvalidated) outcomes.

All patients had at least one dressing applied after surgery, although this varied between adhesive coverings (absorptive or non-absorptive) and tissue adhesive as a dressing. Both the interviews and the analysis of existing RCT outcomes highlighted the practical advantages of dressings (including ability to contain exudate and ease of removal). The interviews also demonstrated that there were psychological factors which affected dressing experience and satisfaction (ie, anxiety about cleanliness of the wound).

Patients with tissue adhesive as a dressing commented that they had been surprised that their wounds had been dressed this way (rather than adhesive dressings which they had had in the past for other wounds). However, these patients stated that compared with their past experiences of adhesive dressings, they liked how glue was transparent, waterproof, did not require multiple applications and came off naturally.

The interviews and the analysis of existing RCT outcomes produced a total of 69 issues. These were grouped into 10 broad categories: wound comfort, exudate and its impact, allergic reactions to the dressing, dressing removal, dressings to protect the wound, impact on daily activities, ease of movement, anxiety about the wound, satisfaction with dressing and wound appearance. Table 2 provides illustrative quotes for the categories identified.

## Phase 2: 'Operationalisation': construction of a provisional measure

A provisional measure was designed based on the findings from phase 1. Nine key categories were included: wound comfort, exudate and its impact, allergic reactions to the dressing, dressing removal, dressings to protect the wound, impact on daily activities, ease of movement, anxiety about the wound and satisfaction with dressing. Issues relating to the appearance of the wound were not included as they were only relevant to longer term outcomes of wound healing (not within first days of surgery). Additionally, since most patients reported having an adhesive dressing, many had not seen their wound within this timeframe. The first version of the measure included 16 items, and was provisionally called the Practical Wound Management Questionnaire.

## Phase 3: Pretesting

Cognitive interviews (n=40) were conducted between July 2015 and March 2016. All interviews were conducted face to face. This consisted of 25 patients who were in hospital and had undergone abdominal general surgery, and 15 healthcare professionals involved in surgical wound care. Demographics are shown in table 1.

Interviews highlighted a number of issues with the measure. For example, items regarding the colour of the wound exudate were removed. Questions were rephrased to focus on the experience of having a

dressing rather than general recovery after surgery (ie, 'Have you been able to perform everyday tasks? (ie, showering/bathing)' was changed to 'Has your dressing prevented you from showering/washing?'). Additionally, since four patients commented that the smell of their wound was missing on the measure, an item was added to capture this.

The measure had intended to be administered 2 days after surgery, although feedback suggested that this needed to be completed up to day 4 as the patient may be disorientated from surgery in the first few days. However, since there were clear differences in recovery with caesarean section and abdominal surgery patients, a timeframe of within 4 days of surgery was set, and the measures recorded the date of surgery and date completed to determine context of responses.

Feedback from patients suggested that it was difficult to respond to questions about exudate, since a healthcare professional cared for their wound while they were in hospital. If their dressing had been changed, they were also uncertain about the reason why (ie, simply as part of standard practice or for other reasons). Therefore, the study team decided to separate the measure into two separate measures. The first related to the practical aspects of wound management and the second related to the patient's experience of the wound/dressing and the psychological aspects (anxiety, satisfaction, etc). The two measures were named the Wound Management Questionnaire and the Wound Experience Questionnaire.

Seven versions of measures were modified throughout the pretesting phase. Pretesting continued until no new issues were identified and no further refinements were believed to be necessary. The final version of The Wound Management Questionnaire contains 4 items, while The Wound Experience Questionnaire contains 10 items online (see supplementary file 5). Overall, the final versions of the measures were well received. In addition, 96% of participants stated that each measure took less than 5 min to complete.

## DISCUSSION

This paper describes the development of two measures for assessing wound management and experience. The Wound Management Questionnaire assesses practical issues early after surgery for completion by healthcare professionals and The Wound Experience Questionnaire assesses patient perceptions of wound healing and satisfaction with their dressings. These measures were developed using a mixed methods approach, including data extraction from 26 published RCTs and interviews with 64 patients and 15 healthcare professionals. Final versions of the measures were easily completed and acceptable to patients and healthcare professionals. Further work is needed to examine their reliability and validity in a wider group of patients.

Given the absence of evidence supporting the effectiveness of dressings for the prevention of SSIs, decision making around dressings needs to be informed by issues such as managing wound exudate, offering physical protection and meeting patients' desires for wound coverage. However, systematic reviews have highlighted a lack of meaningful outcome data on wound symptom management and patient experiences of primary surgical wounds and acceptability of dressings.[4 7 8] To our knowledge, this is the first study to explore these important issues in patients with closed primary surgical wounds. The measures are intended to be used in future studies including a wide variety of primary abdominal wounds such as those created during elective or acute surgery, surgery for benign or malignant disease and bowel resection of obstetric procedures. Such studies will always record the patient group which will be important to consider when looking at the results of the measures.

True patient-centred outcome measures require full consideration of patients' experiences and views.[9 10] The main strength of this research is the use of qualitative research methods to provide important insights into the under researched area of early issues related to primary surgical wounds relating to practical wound management and patient experience. We adopted a purposeful sampling strategy to ensure that perspectives were captured from a range of participants in relation to their primary surgical abdominal wound.[12] Data produced from the interviews were supplemented by an analysis of existing RCT outcomes to ensure that a comprehensive list of issues were initially generated, and therefore acted as a method of triangulation to increase the plausibility and dependability of the interview data.[13]

It is important to note that these measures have only been pretested in relation to primary surgical wounds. Wounds that are intentionally left open or have developed problems are likely to require dressings that have advanced practical properties that are tailored to the wound requirements.[2] Although participants were purposefully sampled, most had had a dressing of some kind (94%). A prospective real-time survey of dressings has demonstrated that this reflects current practice.[17] In addition, these measures have only been pretested in relation to abdominal surgical wounds. However, characteristics of wound healing in this area are likely to be consistent with other parts of the body. Furthermore, these measures focus specifically on the experience of dressings—methods of wound closure (ie, potentially leading to differential ease of removal of sutures or staples) may also affect patient experience, although this would require further investigation.

In summary, our measures can be completed both by patients and by healthcare professionals responsible for postoperative wound care. These measures will now be further developed to ensure that they are appropriate and psychometrically tested instruments, with a view to informing decision making around dressings.

**Acknowledgements** We would like to thank all participants for giving up their time to take part in this study, and members of the Bluebelle study team for their helpful comments on the earlier drafts of the measures.

**Collaborators** The Bluebelle Study Group consists of the following sub-groups: Bluebelle grant co-applicants: Lazaros Andronis (Health Economics Unit, Institute of Applied Health Research, University of Birmingham, Birmingham, UK), Jane Blazeby (Population Health Sciences, Bristol Medical School, University of Bristol, Canynge Hall, 39 Whatley Road, Bristol, UK; University Hospitals Bristol NHS Foundation Trust, Bristol, UK), Natalie Blencowe (Population Health Sciences, Bristol Medical School, University of Bristol, Canynge Hall, 39 Whatley Road, Bristol, UK; University Hospitals Bristol NHS Foundation Trust, Bristol, UK), Melanie Calvert (Institute of Applied Health Research, University of Birmingham, Birmingham, UK), Joanna Coast (Population Health Sciences, Bristol Medical School, University of Bristol, Canynge Hall, 39 Whatley Road, Bristol, UK; NIHR Collaboration for Leadership in Applied Health Research and Care West at University Hospitals Bristol NHS Trust, Bristol, UK), Tim Draycott (North Bristol NHS Trust, Bristol, UK), Jenny L Donovan (Population Health Sciences, Bristol Medical School, University of Bristol, Canynge Hall, 39 Whatley Road, Bristol, UK; NIHR Collaboration for Leadership in Applied Health Research and Care West at University Hospitals Bristol NHS Trust, Bristol, UK), Rachael Gooberman-Hill (Musculoskeletal Research Unit, School of Clinical Sciences, University of Bristol, UK), Robert Longman (University Hospitals Bristol NHS Foundation Trust, Bristol, UK), Laura Magill (Academic Department of Surgery, Queen Elizabeth Hospital, University of Birmingham, UK), Thomas Pinkney (Centre for Patient Reported Outcomes Research, University of Birmingham; Academic Department of Surgery, Queen Elizabeth Hospital, University of Birmingham, UK), Jonathan Mathers (Institute of Applied Health Research, University of Birmingham, Birmingham, UK), Barnaby C Reeves (Clinical Trials and Evaluation Unit, School of Clinical Sciences, University of Bristol, Bristol, UK), Chris A Rogers (Clinical Trials and Evaluation Unit, School of Clinical Sciences, University of Bristol, Bristol, UK), Leila Rooshenas (Population Health Sciences, Bristol Medical School, University of Bristol, Canynge Hall, 39 Whatley Road, Bristol, UK), Andrew Torrance (Institute of Applied Health Research, University of Birmingham, Birmingham, UK), Nicky J Welton (Population Health Sciences, Bristol Medical School, University of Bristol, Canynge Hall, 39 Whatley Road, Bristol, UK), Mark Woodward (University Hospitals Bristol NHS Foundation Trust, Bristol, UK), Trudie Young (Welsh Wound Innovation Centre, Rhodfa Marics, Ynysmaerdy, Pontyclun, Rhondda Cynon Taf, Wales, UK). Other members of the Bluebelle Study Group: Jo Dumville (School of Nursing, Midwifery & Social Work, University of Manchester, Manchester, UK), Daisy Elliott (Population Health Sciences, Bristol Medical School, University of Bristol, Canynge Hall, 39 Whatley Road, Bristol, UK), Louise Flintoff (University Hospitals Bristol NHS Foundation Trust, Bristol, UK), Rhiannon Macefield (Population Health Sciences, Bristol Medical School, University of Bristol, Canynge Hall, 39 Whatley Road, Bristol, UK), David Messenger (University Hospitals Bristol NHS Foundation Trust, Bristol, UK), Christel McMullan (Institute of Applied Health Research, University of Birmingham, Birmingham, UK), Charlotte Murkin (School of Social and Community Medicine, University of Bristol, Bristol, UK; University Hospitals Bristol NHS Foundation Trust, Bristol, UK), Helen van der Nelson (Centre for Patient Reported Outcomes Research, University of Birmingham, UK).

**Contributors** All members of the Bluebelle Study Group read and commented on the final version of the paper. Other roles are described below. Lazaros Andronis co-applicant in the Bluebelle study grant (Health Economics); Jane Blazeby Chief investigator of the Bluebelle study grant (NIHR Bluebelle grant HTA - 12/200/04), led the whole study including chairing of management and executive meetings, supervised development of the questionnaires; Natalie Blencowe co-applicant in the Bluebelle study grant, provided feedback on earlier versions of the questionnaires and recruited patients/health care professionals for interviews; Melanie Calvert co-applicant in the Bluebelle study grant (patient-reported outcomes expertise); Joanna Coast co-applicant in the Bluebelle study grant (Health Economics lead); Jenny L Donovan co-applicant in the Bluebelle study grant (Qualitative research co-lead), co-designed and supervised the qualitative research; Tim Draycott co-applicant in the Bluebelle study grant (Obstetric surgical expertise and leadership) principal investigator and lead advisor for obstetric surgery; Jo Dumville provided feedback on earlier versions of the questionnaires; Daisy Elliott wrote the first draft of the manuscript and edited/finalised the paper, contributed to all aspects of qualitative data collection and qualitative data analysis and led development of the questionnaires; Louise Flintoff recruited patients/health care professionals for interviews; Rachel Gooberman-Hill co-applicant in the Bluebelle study grant (patient and public involvement lead); Robert Longman co-applicant in the Bluebelle study grant (Surgical expertise and leadership); Rhiannon Macefield conducted literature reviews and provided feedback on earlier versions of the questionnaires; Laura Magill co-applicant in the Bluebelle study grant, trial manager in Birmingham; Jonathan Mathers co-applicant in the Bluebelle study grant (Qualitative research co-lead); Christel McMullan contributed to all aspects of qualitative data collection and qualitative data analysis, provided feedback on earlier versions of the questionnaires; David Messenger recruited patients/health care professionals

for interviews; Charlotte Murkin conducted pre-testing interviews; Helen van der Nelson recruited patients/health care professionals for interviews; Tom Pinkney co-applicant in the Bluebelle study grant (Surgical expertise and leadership); Barnaby C Reeves co-applicant in the Bluebelle study grant (Methodological lead), contributed to overall study design, and provided feedback on earlier versions of the questionnaires; Chris A Rogers co-applicant in the Bluebelle study grant (Statistical lead); Leila Rooshenas contributed to all aspects of qualitative data collection and qualitative data analysis in Phase I and provided feedback on earlier versions of the questionnaires; Andrew Torrance co-applicant in the Bluebelle study grant; Nicky J Welton co-applicant in the Bluebelle study extension grant; Mark Woodward co-applicant in the Bluebelle study grant (Paediatric surgical expertise and leadership) and advisor on paediatric surgery; Trudie Young co-applicant in the Bluebelle study grant (Wound nursing specialist) and advisor on wound care.

**Funding** The Bluebelle study (Phase A) is funded by the National Institute for Health Research (NIHR) Health Technology Assessment (HTA) Programme (HTA - 12/200/04) and the MRC ConDuCT-II Hub (Collaboration and innovation for Difficult and Complex randomised controlled Trials In Invasive procedures - MR/K025643/1). JLD and JMB are NIHR Senior Investigators. JLD is also supported by the NIHR Collaboration for Leadership in Applied Health Research and Care (CLAHRC) West at University Hospitals Bristol NHS Foundation Trust.

**Disclaimer** The views expressed are those of the authors and not necessarily those of the MRC, NHS, NIHR or the Department of Health.

**Competing interests** None declared.

**Ethics approval** Ethical approval for this work was granted by the Camden and King's Cross Research Ethics Committee (14/LR/0640) on the 10th April 2014.

**Provenance and peer review** Not commissioned; externally peer reviewed.

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
