## [Reviewer comments · BMJ Open]

ARTICLE DETAILS

TITLE (PROVISIONAL)	Developing outcome measures assessing wound management and patient experience: A mixed methods study
AUTHORS	Elliott, Daisy; Andronis, Lazaros; Blazeby, Jane; Blencowe, Natalie; Calvert, Melanie; Coast, Joanna; Draycott, Tim; Donovan, Jenny; Gooberman-Hill, Rachael; Longman, Robert; Magill, Laura; Mathers, Jonathan; Pinkney, Thomas; Reeves, Barnaby; rogers, chris; Rooshenas, Leila; Torrance, Andrew; Welton, Nicky; Woodward, Mark; Young, Trudie; Dumville, Jo C.; Flintoff, Louise; Macefield, Rhiannon; Messenger, David; McMullan, Christel; Murkin, Charlotte; van der Nelson, Helen

VERSION 1 - REVIEW

REVIEWER	Stefan Acosta Institution of Clinical Sciences, Malmö. Lund University
REVIEW RETURNED	11-Mar-2017

GENERAL COMMENTS	Overall evaluation: This development of methodology in (wound) research is highly appreciated and wanted. However, the rationale to use a mixed sample from abdominal surgery patients and caesarean section patients is not clear. This is a mix of a heterogenous population which should have an impact on the interview results. Whether the patient were operated for a malignant or benign disease, elective or acute should also have a huge impact on the results. Cesarean section is not seldom a problematic wound, especially in the obese. Laparotomy in particular colorectal surgeon patients is often complicated by wound complications. Perhaps this was intentional by the authors do have a diverse expression of results? Can these strengths/limitations be discussed in a separate section in the discussion part. The discussion part is a little bit repetitive from parts from the introduction, methods and results, Specific comments: Abstract-results – “would” should be “wound” Results – Phase1. How many face to face interviews and how many telephone interviews respectively? When in the early postoperative course did these interview take place? (Median, range) to have a better understanding. This should be important for the interview results. There was a statement that most patients had secondary dressings at the time point of interview, which means that the interviews probably took place after a couple of days – before any removal of stitches.
---

REVIEWER	Nana Hyldig and Christina Lindhardt University of Southern Denmark, Odense University Hospital, Denmark
REVIEW RETURNED	20-Mar-2017

GENERAL COMMENTS	It has been a privilege to review this article "Measures for wound management and symptoms of primary surgical wounds". This is a very interesting study in a less researched area within wound management and the study adds value to the research and treatment of wound management. The article is very well written. However, we have a few comments to the article. Abstract Keywords could have been elaborated to enhance the abstract. 5 Uk hospitals, elaborate. Introduction The authors write that the purpose is to develop outcome measures to assess practical wound management issues with closed primary surgical wounds and dressings to use for studies of wound dressings. The authors could elaborate on how clinicians/researchers can benefit from using this instrument in clinical settings and/or studies. Why do the authors solely focus on the use for RCTs? The last paragraph could be rephrased as it needs a few readings to understand the perspective for both patients and healthcare professionals. Methods The bluebell study is mentioned in the first paragraph but not explained before later. Generally the methods section is well written. Under phase 2 and pre testing: Why is cognitive interview used? Page 8, line 12-18: The purpose of the literature search is not clear as the subheading states "literature search to identify existing tools" but in the text it says "to identify RCTs that included outcomes relating to wounds and dressings". We do not think that the use of three systematic reviews should be referred to as a literature search. If the authors had conducted a systematic literature search to identify existing tools, they would have identified tools such as "The Posas scale" (www. Posas.org) and "Wound Registry" (Hollander et al., Wound Registry: Development and Validation) and several others. They should elaborate on how they extract data from the RCTs included in the three reviews. The additional file "Item_trackin_matrix" should be mentioned at page 8, line 25. Results There seem to be confusion in the way the interviews are now described as semi structured. In the section describing the literature search it should be mentioned that the twenty six studies were identified from three reviews as mentioned in the method section. It would have been interesting if
---

	the authors had elaborated on which outcome was most frequently reported in the papers, with reason. Was there any outcomes in the papers that was omitted and why? Page 11, line 7-10: did the patients, that described the benefit of tissue glue as dressing, have earlier experiences with other types of post-surgical dressings? The two questionnaires contains important items to measure patient satisfaction, however, our concern is whether the instrument is sensitive to small changes between two types of dressings. For example in a study conducted by our research group (not yet published) that compared a negative pressure wound therapy dressing with a standard postoperative adhesive dressing, we found that a NPWT dressing statistical significant reduced the pain experience, using a VAS scale. But all the patients had experienced some level of pain within the first 5 days after surgery. So by asking "has the wound been painful", yes/no, we would expect most patients to answer yes if they were asked within the first 4 days post-surgery - regardless of dressing and depending on the type of surgery. The additional file 2, the matrix contains 16 items. The final version of The Wound Management Questionnaire contains four items, while The Wound Experience Questionnaire contains 10 items. A timeframe of within four days of surgery was set. It would help to understand possible applications of the two questionnaires if the final version was illustrated in the end of the results section. Discussion The discussion is stringent and well supported by literature. Page 13 line 43: we think the authors are referring to reference no. 19 instead of reference no. 16.
--	--

REVIEWER	de Vries, Fleur Academic Medical Center Amsterdam, the Netherlands
REVIEW RETURNED	24-Mar-2017

GENERAL COMMENTS	I would like to congratulate the authors with this comprehensive study, the process must have been a lot of work. Although I believe the methodology is good, the manuscript is not well structured. It should be shortened, better structured and terminology should be consistent through the text and checked for spelling mistakes. Also the tables are not numbered well and I do not see the final questionnaires? I have a few other minor comments: General There are abbreviations used in the text which are not explained (eg UK in the abstract). Title The title is quite long. Consider to shorten the title. Abstract There is inconsistent use of terms which makes it confusing
---

	sometimes. Use “practical wound management” , “symptoms associated with” and “patients experience with” and “practical management of primary surgical wounds and patient experience” consistent. Introduction “every year worldwide” should be “worldwide every year”? Page 6, line 5; Why adult surgery? I think it is in every surgery? Methods I am not sure if mentioning who did what is necessary thorough the whole manuscript, e.g. “DE, CMM and LR contacted interested patients to arrange interviews ” is not really relevant. Page 7, line 28: should be elective Page 7, line 30,) is missing 1.2 “three SR were used” is a result, what search was identified to find these SRs? I think the information on the Bluebelle Trial Management group should be in the beginning of the methods. Try to shorten the methods section. Results Try to be consistent in terminology again. The terms semi-structured and face-to-face do not appear in the methods. “would” should be “wound” I guess? “patients were recruited from 5 hospital in the UK” should be in methods. 1.2 where these 26 studies in the 3 systematic reviews? “Attempts to contact the authors” should be in methods. I do not see table 3? I see 2x Table 1? And the last Table has no name? What are the tables at the end of the text? And where is the questionnaire? Discussion The second paragraph is not very relevant. The last sentence can be inserted in the introduction. Last sentence strengths and limitations: “to change” should be removed.
--	--

VERSION 1 – AUTHOR RESPONSE

Reviewer: 1

Reviewer Name: Stefan Acosta

Institution and Country: Institution of Clinical Sciences, Malmö, Lund University

Please state any competing interests or state ‘None declared’: None

Please leave your comments for the authors below:

Overall evaluation: This development of methodology in (wound) research is highly appreciated and wanted. However, the rationale to use a mixed sample from abdominal surgery patients and caesarean section patients is not clear. This is a mix of a heterogenous population which should have

an impact on the interview results. Whether the patient were operated for a malignant or benign disease, elective or acute should also have a huge impact on the results. Cesarean section is not seldom a problematic wound, especially in the obese. Laparotomy in particular colorectal surgeon patients is often complicated by wound complications. Perhaps this was intentional by the authors do have a diverse expression of results? Can these strengths/limitations be discussed in a separate section in the discussion part.

Response: Thank you for your comments. The purpose of the measures being developed is to provide valid assessments of practical wound management issues and patient experience. These issues that are related to the primary wound are generic to abdominal wounds and there is no intention to study the wider (and very important issues) associated with benign/malignant disease or elective or acute surgery. We therefore deliberately included a diverse group of patients to improve the generalisability of the questionnaire to patients undergoing all different types of abdominal surgery with closed primary wounds. In future studies using the new measures, the clinical details of the included groups (e.g. colorectal resection, caesarean section surgery, acute/elective) would always be carefully recorded. This would mean that the outcomes in those new studies will be able to be considered within the context of the patient group being studied. We have now added the following point to the discussion:

“We adopted a purposeful sampling strategy to ensure that perspectives were captured from a range of participants in relation to their primary surgical abdominal wound (10). [...] The measures therefore are intended to be used in future studies including a wide variety of primary abdominal wounds such as those created during elective or acute surgery, surgery for benign or malignant disease and bowel resection of obstetric procedures. Such studies will always record the patient group which will be important to consider when looking at the results of the measures.” (Pages 12 and 13)

The discussion part is a little bit repetitive from parts from the introduction, methods and results,

Response: We have rewritten the discussion to avoid repetition and make it more concise.

Specific comments:

Abstract-results – “would” should be “wound”

Results – Phase1. How many face to face interviews and how many telephone interviews respectively?

When in the early postoperative course did these interview take place? (Median, range) to have a better understanding. This should be important for the interview results. There was a statement that most patients had secondary dressings at the time point of interview, which means that the interviews probably took place after a couple of days – before any removal of stitches.

Response: Thank you for spotting that, the typo has now been corrected. We have also provided additional information about the Phase 1 interviews, including the number of interviews conducted face to face/over the telephone, and patients’ postoperative status:

“A total of 39 interviews were conducted between July 2014 and July 2015. Interviews were conducted in person (n=10), unless patients preferred to be interviewed via telephone (n=29). Interviews lasted an average of 25 minutes (range = 15–50 minutes). The sample consisted of 24 women and 15 men, who mostly described themselves as white British (90%). They had a mean age of 56 years (range 41-88 years). Thirty-seven of the 39 participants had either undergone abdominal general surgery (74%) or a caesarean section (26%), with an average of 18 days since their surgery (range =6-40 days). Two of the 39 patients were scheduled to undergo abdominal general surgery and discussed issues that they anticipated would be important to them.” (Page 8)

Reviewer: 2

Reviewer Name: Nana Hyldig and Christina Lindhardt

Institution and Country: University of Southern Denmark, Odense University Hospital, Denmark

Please state any competing interests or state 'None declared': None declared

Please leave your comments for the authors below:

It has been a privilege to review this article "Measures for wound management and symptoms of primary surgical wounds".

This is a very interesting study in a less researched area within wound management and the study adds value to the research and treatment of wound management. The article is very well written. However, we have a few comments to the article.

Abstract

Keywords could have been elaborated to enhance the abstract.

5 Uk hospitals, elaborate.

Response: Thank you very much for the encouraging feedback. We have added the following keywords: "outcome measures"; "instrument development" and patient-reported outcomes". We have also added more information about the hospitals that participated:

"Two university-teaching NHS hospitals and three district NHS hospitals in the South West and Midlands regions of England." [Abstract, page 3]

Introduction

The authors write that the purpose is to develop outcome measures to assess practical wound management issues with closed primary surgical wounds and dressings to use for studies of wound dressings. The authors could elaborate on how clinicians/researchers can benefit from using this instrument in clinical settings and/or studies. Why do the authors solely focus on the use for RCTs?

Response: We have now provided more information in the introduction about these measures could be used:

"Decision-making around dressings may therefore need to be informed by other properties and qualities that dressings can offer, such as absorption of exudate, patient comfort, offering physical protection, facilitating wound observation, and meeting patients' desires for wound coverage (3). Whilst measures for assessment of wound cosmesis (in the longer term) are available (4, 5), there is a lack of well-developed and validated measurement tools relating to practical wound management or patient experience (2, 3, 6). Such an instrument could be used to monitor the care of individual patients (e.g. assessing the ability of dressings to manage specific symptoms), audits (e.g. quality assurance) and research (e.g. comparing patient satisfaction)." [Page 5]

The last paragraph could be rephrased as it needs a few readings to understand the perspective for both patients and healthcare professionals.

Response: We agree this should have been clearer, and we have now rewritten this paragraph:

"The development of patient-reported outcome measures (PROMs) increasingly includes the use of qualitative research methods that provide the opportunity to elicit and characterise patients' experiences of their health conditions and treatment (7, 8). Qualitative methods can also define health professionals' experiences of care and management (9). Data can be supplemented by expert input and studies in published literature (10, 11). This article describes the development of measures to assess practical wound management issues, symptoms and patient experience associated with primary surgical wounds." [Page 5]

Methods

The bluebell study is mentioned in the first paragraph but not explained before later.

Response: We have now amended this so that the first reference to the Bluebelle study provides an explanation of the research (Page 6).

Generally the methods section is well written. Under phase 2 and pre testing: Why is cognitive interview used?

Response: We have now provided more information about why this technique was used in the pre-testing phase:

"Cognitive interviews are used widely in questionnaire development (10) and involves asking respondents to verbalise their thoughts while answering questions (21). This methodology enabled us to explore the acceptability of the measure and coverage of patients and health care professionals' concerns (in terms of language, accuracy, and relevance) as well as layout (11). During each interview, participants were asked to complete the measure by reading each item aloud and commenting on their understanding. Interviews were guided by a series of probes (e.g. 'What does this item mean to you?', 'Are there other ways you would describe it?'; (21))." [Pages 7 and 8]

Page 8, line 12-18: The purpose of the literature search is not clear as the subheading states "literature search to identify existing tools" but in the text it says "to identify RCTs that included outcomes relating to wounds and dressings". We do not think that the use of three systematic reviews should be referred to as a literature search. If the authors had conducted a systematic literature search to identify existing tools, they would have identified tools such as "The Posas scale" ([www. Posas.org](http://www.Posas.org)) and "Wound Registry" (Hollander et al., Wound Registry: Development and Validation) and several others. They should elaborate on how they extract data from the RCTs included in the three reviews.

Response: We apologise for not being clearer about the methods for this aspect of the study. We have now renamed the subheading as 'Extraction of information from three systematic reviews' (rather than a literature search). We have also provided further information about this in the methods section, and hope this is clearer now. We have also now included the full extraction file (see Additional File 4).

"Three systematic reviews were used to identify RCTs which included outcomes relating to patient experience and management of wound healing. Published papers reporting the studies were obtained where possible. Relevant data were extracted on outcome (as described by the authors), verbatim wording to measure outcome, who reported the outcome, measurement scale and assessment time point. Attempts were made to contact the authors for more information." [Page 7]

The additional file "Item_tracking_matrix" should be mentioned at page 8, line 25.

Response: We have now added this:

"A list of issues from the analysis of the interviews and analysis of existing RCT outcomes was collated into an item tracking matrix, in line with guidance for developing PROMs (19). This is available in the online appendix (see Additional File 3)." [Page 7]

Results

There seem to be confusion in the way the interviews are now described as semi structured.

Response: For consistency, we have removed the reference to "semi-structured" interviews, as we have stated that although there was a topic guide which contained a list of general questions, discussions were ultimately led by the participant (and are thus, semi-structured):

"Interviews were conducted with patients to explore and characterise experiences of wounds and dressings [...] A topic guide was developed (based on literature and views of health care professionals in the Bluebelle study team) to ensure that discussions covered the same core issues but with sufficient flexibility to allow new issues of importance to the informants to emerge." [Page 6]

In the section describing the literature search it should be mentioned that the twenty six studies was identified from three reviews as mentioned in the method section. It would have been interesting if the authors had elaborated on which outcome was most frequently reported in the papers, with reason. Was there any outcomes in the papers that was omitted and why?

Response: We have now stated that the twenty six studies was identified from three reviews, reported which outcome was the most common and included the full data extraction form in a supplementary file (see Additional File 4). We hope that this is clearer now:

“Published papers for twenty six studies that included outcomes relating to patient experience and management of wound healing were identified from the three systematic reviews (22-47). Only two studies included a validated instrument, or modification of a validated instrument, to assess outcomes (24, 38). These were for long term scarring and cosmesis (4, 5). However, no studies reported using validated measures relating to issues associated with practical wound management and patient experiences in the early post-operative period. Descriptions of outcomes were heterogeneous and often poorly defined. The most common reported outcomes related broadly to cosmetic result (reported in 15/26 studies), dressing changes (e.g. frequency, comfort, ease of application and removal; reported in 11/26 studies), and skin reactions (e.g. itching, blistering; reported in 10/26 studies). Full data extraction from the 26 studies is included in Additional File 4.” Page 9]

Page 11, line 7-10: did the patients, that described the benefit of tissue glue as dressing, have earlier experiences with other types of post-surgical dressings?

Response: Patients described their experiences of their current wounds and dressings, but also drew upon their previous experiences. We have now provided additional information about this:

“Patients with tissue glue as a dressing commented that they had been surprised that their wounds had been dressed this way (rather than adhesive dressings which they had had in the past for other wounds). However, these patients stated that compared to past experiences of adhesive dressings, they liked how glue was transparent, waterproof, did not require multiple applications and came off naturally.” (Page 10)

The two questionnaires contains important items to measure patient satisfaction, however, our concern is whether the instrument is sensitive to small changes between two types of dressings. For example in a study conducted by our research group (not yet published) that compared a negative pressure wound therapy dressing with a standard postoperative adhesive dressing, we found that a NPWT dressing statistical significant reduced the pain experience, using a VAS scale. But all the patients had experienced some level of pain within the first 5 days after surgery. So by asking "has the wound been painful", yes/no, we would expect most patients to answer yes if they were asked within the first 4 days post-surgery - regardless of dressing and depending on the type of surgery.

Response: Thank you for sharing your experiences – we agree that the scoring of the measures will require careful consideration and piloting. Currently, our response categories for pain related questions ('Has the wound been painful?', 'Did you feel any pain when the dressing was removed?') are 'Not at all', 'A little', 'Quite a bit', and 'A lot'. We will review this when we psychometrically test the measure which is part of the next phase of work that is planned.

The additional file 2, the matrix contains 16 items. The final version of The Wound Management Questionnaire contains four items, while The Wound Experience Questionnaire contains 10 items. A timeframe of within four days of surgery was set. It would help to understand possible applications of the two questionnaires if the final version was illustrated in the end of the results section.

Response: The questionnaires are now included in the online appendix (see Additional File 5).

Discussion

The discussion is stringent and well supported by literature.

Page 13 line 43: we think the authors are referring to reference no. 19 instead of reference no. 16.
Response: Thank you for spotting this, we have amended this.

Reviewer: 3

Reviewer Name: F. de Vries

Institution and Country: Academic Medical Center Amsterdam, the Netherlands

Please state any competing interests or state 'None declared': None declared

Please leave your comments for the authors below:

I would like to congratulate the authors with this comprehensive study, the process must have been a lot of work. Although I believe the methodology is good, the manuscript is not well structured. It should be shortened, better structured and terminology should be consistent through the text and checked for spelling mistakes. Also the tables are not numbered well and I do not see the final questionnaires?

Response: Thank you for your kind comments. The questionnaires are now included in the online appendix (see Additional File 5).

We have rewritten the introduction and discussion, so that these sections are more concise. However, the detail in the methods section is important to provide a thorough overview of how the questionnaire was developed and pre-tested. The article is under the 4000 word count limit (2735 words in total).

I have a few other minor comments:

General

There are abbreviations used in the text which are not explained (eg UK in the abstract).

Response: Thank you for spotting this. All abbreviations have now been explained.

Title

The title is quite long. Consider to shorten the title.

Response: The title has now been shortened to "Developing outcome measures assessing wound management and patient experience: A mixed methods study".

Abstract

There is inconsistent use of terms which makes it confusing sometimes. Use "practical wound management", "symptoms associated with" and "patients experience with" and "practical management of primary surgical wounds and patient experience" consistent.

Response: We agree we should have been clearer. As suggested, we have now amended this to state that the objective is, "To develop outcome measures to assess practical management of primary surgical wounds and patient experience", and have used consistent terminology throughout the abstract.

Introduction

"every year worldwide" should be "worldwide every year"?

Page 6, line 5; Why adult surgery? I think it is in every surgery?

Response: We have updated "every year worldwide" to "worldwide every year". We referred to adult surgery specifically because a recent study by the Bluebelle team has shown that paediatric professionals report that they many do not routinely apply dressings to primary wounds (Rooshenas et al, 2016).

Methods

I am not sure if mentioning who did what is necessary through the whole manuscript, e.g. “DE, CMM and LR contacted interested patients to arrange interviews” is not really relevant.

Page 7, line 28: should be elective

Page 7, line 30,) is missing

1.2 “three SR were used” is a result, what search was identified to find these SRs?

I think the information on the Bluebelle Trial Management group should be in the beginning of the methods.

Try to shorten the methods section.

Response: We have now removed specific references to the researchers, and have removed these errors in the text – thank you for spotting these. The term ‘literature search’ is confusing, so we have now referred to this as ‘data extraction from systematic reviews’, we hope this is clearer. We appreciate that the methods section is long, although feel that this level of detail is required to provide a transparent and thorough explanation of the three phases of data collection. Furthermore, the nature of the qualitative research means that it generates more words than a quantitative paper and is less readily summarised in tabular form. The article is under the 4000 word count limit (2735 words in total).

Results

Try to be consistent in terminology again. The terms semi-structured and face-to-face do not appear in the methods.

Response: We agree this should be more consistent. For consistency, we have removed the reference to “semi-structured” interviews, as we have later stated that although there was a topic guide which contained a list of general questions but discussions were ultimately led by the participant. In Phase 3, a particular type of interviews were conducted (cognitive interviews), and we have now included an explanation of what these interviews involved (Page 7/8). In the Phase 1 and 3 results sections, we have provided specific information about the way that these interviews were conducted (face to face or over the telephone; pages 8 and 11). In all other instances throughout the text, we have now referred to ‘interviews’.

“would” should be “wound” I guess?

“patients were recruited from 5 hospital in the UK” should be in methods.

1.2 where these 26 studies in the 3 systematic reviews?

“Attempts to contact the authors” should be in methods.

I do not see table 3? I see 2x Table 1? And the last Table has no name?

What are the tables at the end of the text? And where is the questionnaire?

Response: We have now corrected the typo. We have removed references to where participants were recruited from the results section, and have provided this information in the methods section. We have added that “Published papers for twenty six studies that included outcomes relating to patient experience and management of wound healing were identified from the three systematic reviews (22-47)” (Page 9). The sentence on attempts to contact the authors has also been moved to the methods section.

We apologise for the confusion with tables. The COREQ table was listed as Table 1 as it was included as a separate document to the manuscript (and is referenced in the text as Additional File 1). Similarly, the item tracking matrix is in Additional File 2 (we have now added a caption to this). The questionnaires are now included in the online appendix (see Additional File 5).

Discussion

The second paragraph is not very relevant. The last sentence can be inserted in the introduction.

Last sentence strengths and limitations: "to change" should be removed.

Response: We have now deleted the second paragraph, and rewritten the discussion. We have also removed "to change" in the strengths and limitations section.

VERSION 2 – REVIEW

REVIEWER	Nana Hyldig and Christina Lindhard University of Southern Denmark, Odense University Hospital, Denmark
REVIEW RETURNED	25-May-2017

GENERAL COMMENTS	We congratulate the authors on a successfully completed correction of their manuscript and recognize the authors for responding to each issue raised by the three reviewers. Nevertheless, we do still have some issues with the section on data extraction from existing literature. We recognize that the authors have done a great job with extraction of information from 26 RCT. However, the background for selecting the three specific systematic reviews (SRs) is still not clarified. 1) Please elaborate on how and why the authors identify and select these three SRs in the method section. Please add the references of the SRs (page 8, l 9). 2) Without clarification of the reason for selecting these particular SRs, it leaves us wondering why the authors do not conduct a systematic literature search in order to identify other studies with better-defined outcomes: a) In the result section the authors write that the descriptions of outcomes were heterogeneous and often poorly defined in the RCTs. b) The first author of three SRs mention in the discussion (p 13, l 53) is also co-authors on this paper. We do not expect the authors to redo their literature search, but we believe the reason for using RCTs included in the three SRs (to identify outcomes in the literature) would be more transparent if the authors elaborate on the reason for selecting the SRs.
--

REVIEWER	Fleur de Vries Academic Medical Centre Amsterdam
REVIEW RETURNED	07-Jun-2017

GENERAL COMMENTS	The manuscript has been improved and is well structured now.
--

VERSION 2 – AUTHOR RESPONSE

We are very pleased to see that the changes to the paper were well-received and we appreciate the reviewers' helpful and constructive comments. We have listed our response to each point raised by the reviewers and highlight the changes made to the manuscript. We look forward to your response.

Comment: Please elaborate on how and why the authors identify and select these three SRs in the method section. Please add the references of the SRs (page 8, l 9).

Response: We purposefully selected the three SRs and two other RCTs to inform these measures rather than conducting a formal SR. We did this because we thought that the trials included within these SRs and the additional two RCTs would provide sufficient coverage of the relevant literature to inform our work. The specific reasons for the purposeful selection are outlined below.

- The Cochrane review of dressings for the prevention of SSI (Dumville et al, 2011). This review was included because it was the review that informed the commission call for the Bluebelle study (The Bluebelle study: a feasibility study of three wound dressing strategies in elective and unplanned surgery, HTA - 12/200/04). This work was funded within this grant and the Cochrane review was well conducted and very comprehensive.
- The Chow et al (2010) review and the Dumville et al (2014a) review. These reviews were selected because they included papers that had studied tissue glue as-a-method of wound closure (not as-a-dressing). In the Bluebelle feasibility study it was identified that tissue glue was being used-as-a dressing and we wanted to look for outcomes relevant to that intervention. These reviews were included because they were the only reviews we could find at the time that had comprehensively included RCTs using tissue glue in the surgical wound.
- Two additional RCTs (Burke 2012 and Ravskog 2011) were included. We added these because they were identified from the on-going update of the Dumville et al (2014b) review of dressings in which we became involved.

The Supplementary File 4 gives a clear indication of the RCTs that were included and their source (coded as '4=additional studies provided by authors of the Cochrane dressings review update').

Taken together, whilst this was not a comprehensive search, we considered that the RCTs and reviews included outcomes relevant to dressings, no dressing and tissue glue. We hope that this is clearer now, and apologise for the confusion. We have updated the methods section so that it reads:

“Systematic reviews were purposefully selected to identify RCTs measuring outcomes relating to patient experience and management of wound healing. Since the wider Bluebelle study was exploring whether a trial comparing different types of dressings (including dressings versus the novel use of tissue glue as a dressing) to no dressing was possible, we selected three recent systematic reviews (4, 20, 21) to identify RCTs which included outcomes relevant to both dressings and the use of tissue adhesive. Although not published at the time of conducting this work, additional references from an updated version of one of the systematic reviews (7) were also provided. Published papers reporting the RCTs included in the systematic reviews were obtained where possible. Relevant data from the RCT reports were then extracted on the outcome (as described by the authors), the verbatim wording to measure outcome, who reported the outcome, the measurement scale and the assessment time point. Attempts were made to contact the authors for more information.” (Page 7)

Comment: Without clarification of the reason for selecting these particular SRs, it leaves us wondering why the authors do not conduct a systematic literature search in order to identify other studies with better-defined outcomes: a) In the result section the authors write that the descriptions of outcomes were heterogeneous and often poorly defined in the RCTs.

Response: We chose to examine outcomes in RCTs identified by existing systematic reviews (as described above). We did not intend to pre-specify to choose studies only with well-defined outcomes in order to capture all information on the outcomes that were measured regardless of the quality of the reporting.

Comment: The first author of three SRs mention in the discussion (p 13, l 53) is also co-authors on this paper. We do not expect the authors to redo their literature search, but we believe the reason for using RCTs included in the three SRs (to identify outcomes in the literature) would be more transparent if the authors elaborate on the reason for selecting the SRs.

Response: It is correct that Jo Dumville is co-author on this paper. She joined the Bluebelle team one year after the grant opened, and was not involved in the initial selection of the systematic reviews. As described above, we have now added more information to the manuscript and hope this is clearer now.

References:

Dumville JC, Walter CJ, Sharp CA, Page T. (2011). Dressings for the prevention of surgical site infection. *Cochrane Database of Systematic Reviews*, 7, CD003091.

Dumville JC, Coulthard P, Worthington HV, et al. (2014a). Tissue adhesives for closure of surgical incisions. *Cochrane Database of Systematic Reviews*, 28(11): CD004287.

Chow A, Marshall H, Zacharakis E, Paraskeva P, Purkayastha S. (2010). Use of tissue glue for surgical incision closure: a systematic review and meta-analysis of randomized controlled trials. *Journal of the American College of Surgeons*, 211(1), 114-25.

Dumville JC, Gray TA, Walter CJ, Sharp CA, Page T. (2014b). Dressings for the prevention of surgical site infection. *Cochrane Database of Systematic Reviews*, 1, CD003091.